# Biases for Emergent Communication in Multi-agent Reinforcement Learning

**Tom Eccles**
DeepMind
London, UK
eccles@google.com

**Yoram Bachrach**
DeepMind
London, UK
yorambac@google.com

**Guy Lever**
DeepMind
London, UK
guylever@google.com

**Angeliki Lazaridou**
DeepMind
London, UK
angeliki@google.com

**Thore Graepel**
DeepMind
London, UK
thore@google.com

## Abstract

We study the problem of emergent communication, in which language arises because speakers and listeners must communicate information in order to solve tasks. In temporally extended reinforcement learning domains, it has proved hard to learn such communication without centralized training of agents, due in part to a difficult joint exploration problem. We introduce inductive biases for positive signalling and positive listening, which ease this problem. In a simple one-step environment, we demonstrate how these biases ease the learning problem. We also apply our methods to a more extended environment, showing that agents with these inductive biases achieve better performance, and analyse the resulting communication protocols.

## 1   Introduction

Environments where multiple learning agents interact can model important real-world problems, ranging from multi-robot or autonomous vehicle control to societal social dilemmas [4, 26, 22]. Further, such systems leverage implicit natural curricula, and can serve as building blocks in the route for constructing aritficial general intelligence [21, 1]. Multi-agent games provide longstanding grand-challenges for AI [16], with important recent successes such as learning a cooperative and competitive multi-player first-person video game to human level [14]. An important unsolved problem in multi-agent reinforcement learning (MARL) is communication between independent agents. In many domains, agents can benefit from sharing information about their beliefs, preferences and intents with their peers, allowing them to coordinate joint plans or jointly optimize objectives.

A natural question that arises when agents inhibiting the same environment are given a communication channel without an agreed protocol of communication is that of emergent communication [32, 35, 8]: how would the agents learn a "language" over the joint channel, allowing them to maximize their utility? The most naturalistic model for emergent communication in MARL is that used in Reinforced Inter-Agent Learning (RIAL) [8] where agents optimize a message policy via reinforcement from the environment's reward signal. Unfortunately, straightforward implementations perform poorly [8],

driving recent research to focus on differentiable communication models [8, 29, 33, 12], even though these models are less generally applicable or realistic.

RIAL offers the advantage of having *decentralized* training and execution; similarly to human communication, each agent treats others as a part of its environment, without the need to have access to other agents' internal parameters or to back-propagate gradients "through" parameters of others. Further, agents communicate with *discrete* symbols, providing symbolic scaffolding for extending to natural language. We build on these advantages, while facilitating joint exploration and learning via communication-specific inductive biases.

We tackle emergent communication through the lens of Paul Grice [10, 30], and capitalize on the dual view of communication in which interaction takes place between a *speaker*, whose goal is to be informative and relevant (adhering to the equivalent Gricean maxims), and a *listener*, who receives a piece of information and assumes that their speaker is cooperative (providing informative and relevant information). Our methodology is inspired by the recent work of Lowe et al. [24], who proposed a set of comprehensive measures of emergent communication along two axis of *positive signalling* and *positive listening*, aiming at identifying real cases of communication from pathological ones.

**Our contribution:** we formulate losses which encourage positive signaling and positive listening, which are used as auxiliary speaker and listener losses, respectively, and are appended to the RIAL communication framework. We design measures in the spirit of Lowe et al. [24] but rather than using these as an introspection tool, we use them as an optimization objective for emergent communication. We design two sets of experiments that help us clearly isolate the real contribution of communication in task success. In a one-step environment based on summing MNIST digits, we show that the biases we use facilitate the emergence of communication, and analyze how they change the learning problem. In a gridworld environment based on search of a treasure, we show that the biases we use make communication appear more consistently, and we interpret the resulting protocol.

## 1.1 Related Work

Differentiable communication was considered for discrete messages [8, 29] and continuous messages [33, 12], by allowing gradients to flow through the communication channel. This improves performance, but effectively models multiple agents as a single entity. In contrast we assume agents are *independent* learners, making the communication channel non-differentiable. Earlier work on emergent communication focused on cooperative "embodied" agents, showing how communication helps accomplish a common goal [8, 29, 6], or investigating communication in mixed cooperative-competitive environments [25, 3, 15], studying properties of the emergent protocols [20, 17, 24].

Previous research has investigated independent reinforcement learners in cooperative settings [34], with more recent work focusing on canonical RL algorithms. One version of decentralized Q-learning converges to optimal policies in deterministic tabular environments without additional communication [18], but does not trivially extend to stochastic environments or function approximation. Centralized critics [25, 9] improve stability by allowing agents to use information from other agents during training, but these violate our assumptions of independence, and may not scale well.

## 2 Setting

We apply **multi-agent reinforcement learning** (MARL) in partially-observable Markov games (i.e. *partially-observable stochastic games*) [31, 23, 11], in environments where agents have a joint communication channel. In every state, agents take actions given partial observations of the true world state, including messages sent on a shared channel, and each agent obtains an individual reward. Through their individual experiences interacting with one another and the environment, agents learn to broadcast appropriate messages, interpret messages received from peers and act accordingly.

Formally, we consider an $N$-player partially observable Markov game $G$ [31, 23] defined on a finite state set $\mathcal{S}$, with action sets $(\mathcal{A}^1, \ldots, \mathcal{A}^N)$ and message sets $(\mathcal{M}^1, \ldots, \mathcal{M}^N)$. An observation function $O : \mathcal{S} \times \{1, \ldots, N\} \to \mathbb{R}^d$ defines each agent's $d$-dimensional restricted view of the true state space. On each timestep $t$, each agent $i$ receives as an observation $o_t^i = O(\mathcal{S}_t, i)$, and the messages $m_{t-1}^j$ sent in the previous state for all $j \neq i$. Each agent $i$ then select an environment action $a_t^i \in \mathcal{A}^i$ and a message action $m_t^i \in \mathcal{M}^i$. Given the joint action $(a_t^1, \ldots, a_t^N) \in (\mathcal{A}^1, \ldots, \mathcal{A}^N)$ the state changes based on a transition function $\mathcal{T} : \mathcal{S} \times \mathcal{A}^1 \times \cdots \times \mathcal{A}^N \to \Delta(\mathcal{S})$; this is a stochastic transition, and we denote the set of discrete probability distributions over $\mathcal{S}$ as $\Delta(\mathcal{S})$. Every

agent gets an individual reward $r_t^i : \mathcal{S} \times \mathcal{A}^1 \times \cdots \times \mathcal{A}^N \to \mathbb{R}$ for player $i$. We use the notation $\mathbf{a_t} = (a_t^1, \ldots, a_t^N)$, $\mathbf{m_t} = (m_t^1, \ldots, m_t^N)$, and $\mathbf{o_t} = (o_t^1, \ldots o_t^N)$. We write $\mathbf{m}_{\bar{i},t}$ for $(m_t^1, \ldots, m_t^N)$, excluding $m_t^i$, and $\boldsymbol{\mathcal{M}}_{\bar{i}}$ for $(\mathcal{M}^1, \ldots, \mathcal{M}^N)$, excluding $\mathcal{M}^i$.

In our fully cooperative setting, each agent receives the same reward at each timestep, $r_t^i = r_t^j \quad \forall i, j \leq N$, which we denote by $r_t$. Each agent maintains an action and a message policy from which actions and messages are sampled, $a_t^i \sim \pi_A^i(\cdot|x_t^i)$ and $m_t^i \sim \pi_M^i(\cdot|x_t^i)$, and which can in general be functions of their entire trajectory of experience $x_t^i := (\mathbf{m}_0, o_1^i, a_1^i, \ldots, a_{t-1}^i, \mathbf{m}_{t-1}, o_t^i)$. These policies are optimized to maximize discounted cumulative joint reward $J(\boldsymbol{\pi}_A, \boldsymbol{\pi}_M) := \mathbb{E}_{\boldsymbol{\pi}_A, \boldsymbol{\pi}_M, \mathcal{T}} \left[ \sum_{t=1}^{\infty} \gamma^{t-1} r_t \right]$ (which is discounted by $\gamma < 1$ to ensure convergence), where $\boldsymbol{\pi}_A := \{\pi_A^1, \ldots, \pi_A^N\}, \boldsymbol{\pi}_M := \{\pi_M^1, \ldots, \pi_M^N\}$. Although the objective $J(\boldsymbol{\pi}_A, \boldsymbol{\pi}_M)$ is a joint objective, our model is that of decentralized learning and execution, where every agent has its own experience in the environment, and independently optimizes the objective $J$ with respect to its own action and message policies $\pi_A^i$ and $\pi_M^i$; there is no communication between agents other than using the actions and message channel in the environment. Applying independent reinforcement learning to cooperative Markov games results in a problem for each agent which is non-stationary and non-Markov, and presents difficult joint exploration and coordination problems [2, 5, 19, 27].

# 3   Shaping Losses for Facilitating Communication

One difficulty in emergent communication is getting the communication channel to help with the task at all. There is an equilibrium where the speaker produces random symbols, and the listener's policy is independent of the communication. This might seem like an unstable equilibrium: if one agent uses the communication channel, however weakly, the other will have some learning signal. However, this is not the case in some tasks. If the task without communication has a single, deterministic optimal policy, then messages from policies sufficiently close to the uniform message policy should be ignored by the listener. Furthermore, any entropy costs imposed on the communication channel, which are often crucial for exploration, exacerbate the problem, as they produce a positive pressure for the speaker's policy to be close to random. Empirically, we often see agents fail to use the communication channel at all; but when agents start to use the channel meaningfully, they are then able to find at least a locally optimal solution to the communication problem.

We propose two shaping losses for communication to alleviate these problems. The first is for *positive signalling* [24]: encouraging the speaker to produce diverse messages in different situations. The second is for *positive listening* [24]: encouraging the listener to act differently for different messages. In each case, the goal is for one agent to learn to ascribe some meaning to the communication, even while the other does not, which eases the exploration problem for the other agent.

We note that most policies which maximize these biases do not lead to high reward. Much information about an agent's state is unhelpful to the task at hand, so with a limited communication channel positive signalling is not sufficient to have useful communication. For positive listening, the situation is even worse – most ways of conditioning actions on messages are actively unhelpful to the task, particularly when the speaker has not developed a good protocol. These losses should therefore not be expected to lead directly to good communication. Rather, they are intended to ensure that the agents begin to use the communication channel at all – after this, MARL can find a useful protocol.

## 3.1   Bias for positive signalling

The first inductive bias we use promotes positive signalling, incentivizing the speaker to produce different messages in different situations. We add a loss term which is minimized by message policies that have high mutual information with the speaker's trajectory. This encourages the speaker to produce messages uniformly at random overall, but non-randomly when conditioned on the speaker's trajectory.

We denote by $\overline{\pi_M^i}$ the average message policy for agent $i$ over all trajectories, weighted by how often they are visited under the current action policies for all agents. The mutual information of agent $i$'s message $m_t^i$ with their trajectory $x_t^i$ is:

$$\mathcal{I}(m_t^i, x_t^i) = \mathcal{H}(m_t^i) - \mathcal{H}(m_t^i | x_t^i) \tag{1}$$

$$= -\sum_{m \in \mathcal{M}_i} \overline{\pi_M^i}(m) \log(\overline{\pi_M^i}(m)) + \mathbb{E}_{x_t^i} \sum_{m \in \mathcal{M}_i} \pi_M^i(m|x_t^i) \log(\pi_M^i(m|x_t^i)) \tag{2}$$

We estimate this mutual information from a batch of rollouts of the agent policy. We calculate $\mathcal{H}(m_t^i|x_t^i)$ exactly for each timestep from the agent's policy. To estimate $\mathcal{H}(\overline{\pi_m})$, we estimate $\overline{\pi_m}$ as the average message policy in the batch of experience. Intuitively, we would like to maximize $\mathcal{I}(m_t^i, x_t^i)$, so that the speaker's message depends maximally on their current trajectory. However, adding this objective as a loss for gradient descent leads to poor solutions. We hypothesize that this is due to properties of the loss landscape for this loss. Policies which maximize mutual information are deterministic for any particular trajectory $x_t^i$, but uniformly random unconditional on $x_t^i$. At such policies, the gradient of the term $\mathcal{H}(\pi_M^i(\cdot|x_t^i))$ is infinite. Further, for any $c < \log(2)$ the space of policies which have entropy at most $c$ is disconnected, in that there is no continuous path in policy space between some policies in this set.

To overcome these problems, we instead use a loss which is minimized for a high value for $\mathcal{H}(\overline{\pi_M^i})$ and a target value for $\mathcal{H}(\pi_M^i|s_i)$. The loss we use is:

$$L_{ps}(\pi_M^i, s_i) = -\mathbb{E}\big(\lambda\mathcal{H}(\overline{\pi_M^i}) - (\mathcal{H}(m_t^i|x_t) - \mathcal{H}_{target})^2\big), \qquad (3)$$

for some target entropy $\mathcal{H}_{target}$, which is a hyperparameter. This loss has finite gradients around its minima, and for suitable choices of $\mathcal{H}_{target}$ the space of policies which minimizes this loss is connected. In practice, we found low sensitivity to $\mathcal{H}_{target}$, and typically use a value of around $\log(|A|)/2$, which is half the maximum possible entropy.

---

**Algorithm 1** Calculation of positive signalling loss

---

1: $\overline{\pi_M}_i \leftarrow 0$.
2: $L_{ps} \leftarrow 0$.
3: Target conditional entropy $\mathcal{H}_{target}$.
4: Weighting $\lambda$ for conditional entropy.
5: **for** b=1; b $\leq$ B; b++ **do** # Batch of rollouts.
6:     Observations $o_t^i$ for $1 \leq t \leq T$.
7:     Actions $a_t^i$ for $1 \leq t \leq T$.
8:     Other agent messages $\mathbf{m}_{t,\bar{i}}$ from $\boldsymbol{\mathcal{M}}_{\bar{i}}$ for $1 \leq t \leq T$.
9:     Initial hidden state $h_0^i$.
10:     Action set $\mathcal{A}^i$, message set $\mathcal{M}^i$, observation space $\mathcal{O}^i$, hidden state space $H^i$.
11:     Message policy $\pi_M^i : \mathcal{O}^i \times \mathcal{A}^i \times H^i \times \boldsymbol{\mathcal{M}}_{\bar{i}} \mapsto \mathcal{M}^i$.
12:     Hidden state update rule $h^i : \mathcal{O}^i \times \mathcal{A}^i \times H^i \times \boldsymbol{\mathcal{M}}_{\bar{i}} \mapsto H^i$.
13:     **for** t = 1; t $\leq$ T; t++ **do**
14:         $h_t^i \leftarrow h^i(o_t^i, a_{t-1}^i, h_{t-1}^i, \mathbf{m}_{t-1,\bar{i}})$.
15:         $p_t^i \leftarrow \pi_M^i(o_t^i, h_{t-1}^i, \mathbf{m}_{t-1,\bar{i}})$.
16:         $\overline{\pi_M} \leftarrow \overline{\pi_M} + \pi_t^i/(T \times B)$.
17:         $\mathcal{H}_t^i \leftarrow \sum_{m \in \mathcal{M}^i} p_t^i(m) \log(p_t^i(m))$.
18:         $L_{ps} \leftarrow L_{ps} + \lambda(\mathcal{H}_t^i - \mathcal{H}_{target})^2$.
19:     **end for**
20: **end for**
21: $\mathcal{H} = \sum_{m \in \mathcal{M}^i} \overline{\pi}(m) \log(\overline{\pi}(m))$.
22: $L_{ps} \leftarrow L_{ps} + T \times B \times \mathcal{H}$.

---

### 3.2 Bias for positive listening

The second bias promotes positive listening: encouraging the listener to condition their actions on the communication channel. This gives the speaker some signal to learn to communicate, as its messages have an effect on the listener's policy and thus on the speaker's reward. The way we encourage positive listening is akin to the *causal influence of communication*, or CIC [15, 24]. In [15], this was used as a bias for the speaker, to produce influential messages, and in [24] as a measure of whether communication is taking place. We use a similar measure *as a loss* for the listener to be influenced by messages. In [15, 24], CIC was defined over one timestep as the mutual information between the speaker's message and the listener's action. We extend this to multiple timesteps using the mutual information between all of the speaker's previous messages on a single listener action – using this as an objective encourages the listener to pay attention to all the speaker's messages, rather than just the most recet. For a listener trajectory $x_t = (o_1, a_1, \mathbf{m}_1, \ldots, o_{t-1}, a_{t-1}, \mathbf{m}_{t-1}, o_t)$, we define

$x'_t = (o_1, a_1, \ldots, a_{t-1}, o_t)$ (this is the trajectory $x_t$, with the messages removed). We define the multiple timestep CIC as:

$$CIC(x_t) = \mathcal{H}(a_t|x'_t) - \mathcal{H}(a_t|x_t). \tag{4}$$

$$= D_{KL}\big((a_t|x_t)||(a_t|x'_t)\big) \tag{5}$$

We estimate this multiple timestep CIC by learning the distribution $\pi^i_A(\cdot|x'_t)$. We do this by performing a rollout of the agent's policy network, with the actual observations and actions in the trajectory, and zero inputs in the place of the messages. We fit the resulting function $\overline{\pi}^i_A(\cdot|x'_t)$ to predict $\pi^i_A(\cdot|x_t)$, using a cross-entropy loss between these distributions:

$$L_{ce}(x_t) = - \sum_{a \in \mathcal{A}^i} \pi^i_A(a|x_t) \log(\overline{\pi}^i_A(a|x'_t)), \tag{6}$$

where we backpropagate only through the $\overline{\pi}^i_A(a|x_t)$ term. For a given policy $\pi^i_A$, this loss is minimized in expectation when $\overline{\pi}^i_A(\cdot|x'_t) = \mathbb{E}(\pi^i_A(\cdot|x'_t))$. Thus $\overline{\pi}^i_A$ is trained to be an approximation of the listener's policy unconditioned on the messages it has received. The multi-timestep CIC can then be estimated by the KL divergence between the message-conditioned policy and the unconditioned policy:

$$CIC(x_t) \approx D_{KL}(\pi^i_A(\cdot|x_t)||\overline{\pi}^i_A(\cdot|x'_t)). \tag{7}$$

For training positive listening we use a different divergence between these two distributions, which we empirically find achieves more stable training. We use the $L_1$ norm between the two distributions:

$$L_{p_l}(x_t) = - \sum_{a \in \mathcal{A}^i} \big|\pi^i_A(a|x_t) - \overline{\pi}^i_A(a|x'_t)\big|. \tag{8}$$

---

**Algorithm 2** Calculation of positive listening losses

1: Observations $o^i_t$ for $1 \leq t \leq T$.
2: Actions $a^i_t$ for $1 \leq t \leq T$.
3: Other agent messages $\mathbf{m}_{t,\overline{i}}$ from $\boldsymbol{\mathcal{M}}_{\overline{i}}$ for $1 \leq t \leq T$.
4: Initial hidden state $h^i_0 = h'_0$.
5: Action set $\mathcal{A}^i$, observation space $\mathcal{O}^i$, hidden state space $H^i$.
6: Action policy $\pi^i_A : \mathcal{O}^i \times \mathcal{A}^i \times H^i \times \boldsymbol{\mathcal{M}}_{\overline{i}} \mapsto \mathcal{A}^i$.
7: Hidden state update rule $h^i : \mathcal{O}^i \times \mathcal{A}^i \times H^i \times \boldsymbol{\mathcal{M}}_{\overline{i}} \mapsto H^i$.
8: $L_{ce} \leftarrow 0$.
9: $L_{pl} \leftarrow 0$.
10: **for** i = 1; t $\leq$ T; t++ **do**
11: $\quad h^i_t \leftarrow h^{\overline{i}}(o^i_t, a^i_{t-1}, h^i_{t-1}, \mathbf{m}_{t-1,\overline{i}})$.
12: $\quad p^i_t \leftarrow \pi^i_A(o^i_t, h^i_{t-1}, \mathbf{m}_{t-1,\overline{i}})$.
13: $\quad h'_t \leftarrow h^i(o^i_t, a^i_{t-1}, h'_{t-1}, \mathbf{0})$.
14: $\quad \overline{p}^i_t \leftarrow \pi^i_A(o^i_t, a^i_{t-1}, h'_t, \mathbf{0})$.
15: $\quad L_{ce} \leftarrow L_{ce} + \sum_{a \in A} stop\_gradient(p^i_t(a)) \log(\overline{p}^i_t(a))$.
16: $\quad L_{pl} \leftarrow L_{pl} + \sum_{a \in A} |p^i_t(a) - stop\_gradient(\overline{p}^i_t(a))|$.
17: **end for**

---

## 4  Empirical Analysis

We consider two environments. The first is a simple one-step environment, where agents must **sum MNIST digits** by communicating their value. This environment has the advantage of being very amenable to analysis, as we can readily quantify how valuable the communication channel currently is to each agent. In this environment, we provide evidence for our hypotheses about *how* the biases we introduce in Section 3 ease the learning of communication protocols. The second environment is a new multi-step MARL environment which we name **Treasure Hunt**. It is designed to have a clear

performance ceiling for agents which do not utilise a communication channel. In this environment, we show that the biases enable agents to learn to communicate in a multi-step reinforcement learning environment. We also analyze the resulting protocols, finding interpretable protocols that allow us to intervene in the environment and observe the effect on listener behaviour. The full details of the Treasure Hunt environment, together with the hyperparameters used in our agents, can be found in the supplementary material.

## 4.1 Summing MNIST digits

In this task, depicted in Figure 1, the speaker and listener agents each observe a different MNIST digit (as an image), and must determine the sum of the digits. The speaker observes an MNIST digit, and selects one of 20 possible messages. The listener receives this message, observes an independent MNIST digit, and must produce one of 19 possible actions. If this action matches the sum of the digits, both agents get a fixed reward of 1, otherwise, both receive no reward. The agents used in this environment consist of a convolutional neural net, followed by an multi-layer perceptron and a linear layer to produce policy logits. For the listener, we concatenate the message sent to the output of the convnet as a one-hot vector. The agents are trained independently with REINFORCE.

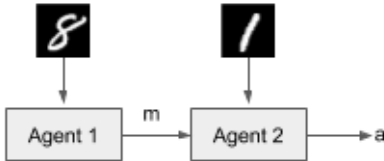

Figure 1: Summing MNIST environment. In this example, both agents would get reward 1 if $a = 9$.

The purpose of this environment is to test whether and how the biases we propose ease the learning task. To do this, we quantify how useful the communication channel is to the speaker and to the listener. We periodically calculate the rewards for the following policies:

1. The optimal listener policy $\pi_{l_c}$, given the current speaker and the labels of the listener's MNIST digits.

2. The optimal listener policy $\pi_{l_{nc}}$, given the labels of the listener's MNIST digits and no communication channel.

3. The optimal speaker policy $\pi_{s_c}$, given the current listener and the labels of the speaker's MNIST digits.

4. The uniform speaker policy $\pi_{s_u}$, given the current listener.

We calculate these quantities by running over many batches of MNIST digits, and calculating the optimal policies explicitly. The reward the listener can gain from using the communication channel is $P_l(\pi_s) = R(\pi_{l_c}, \pi_s) - R(\pi_{l_{nc}}, \pi_s)$, so this is a proxy for the strength of the learning signal for the listener to use the channel. Similarly, $P_s(\pi_s) = R(\pi_l, \pi_{s_c}) - R(\pi_l, \pi_{s_u})$ is how much reward the speaker can gain from using the communication channel, and so is a proxy for the strength of the learning signal for the speaker.

The results (Figure 2) support the hypothesis that the bias for positive signalling eases the learning problem for the listener, and the bias for positive listening eases the learning problem for the speaker. When neither agent has any inductive bias, we see both $P_l$ and $P_s$ stay low throughout training, and the final reward of 0.1 is exactly what can be achieved in this environment with no communication. When we add a bias for positive signalling or positive listening, we see the communication channel used in most runs (Table 1), leading to greater reward, and $P_s$ and $P_l$ both increase. Importantly, when we add our inductive bias for positive listening, we see $P_s$ increase initially, followed by $P_l$. This is consistent with the hypothesis that the positive listening bias bias produces a stronger learning signal for the speaker; then once the speaker has begun to learn to communicate meaningfully, the listener also has a strong learning signal. When we add the bias for positive signalling the reverse is true – $P_l$ increases before $P_s$. This again fits the hypothesis that speaker's bias produces a stronger learning signal for the speaker.

We also ran experiments with the speaker getting an extra reward for positive listening, as in [15]. However, we did not see any gain from this over the no-bias baseline; in our setup, it seems the

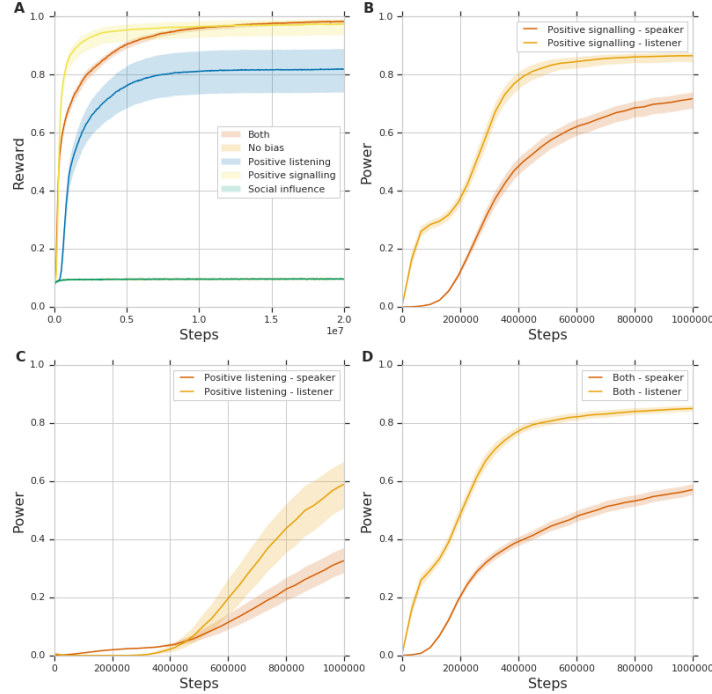

Figure 2: (a) Both biases lead to more reward. (b,c, d) Listener and speaker power in various settings. Listener power increases first with positive signalling, and speaker power increases first with positive listening.

speaker agent was unable to gain any influence over the listener. We think that there is a natural reason this bias would not help in this environment; for a fixed listener, the speaker policy which optimizes positive listening has no relation to the speaker's input. Thus this bias does not force the speaker to produce different messages for different inputs, and so does not increase the learning signal for the listener.

| Biases | Proportion of good runs | CI | Final reward of good runs |
|---|---|---|---|
| No bias | 0.00 | 0.00-0.07 | N/A |
| Social influence | 0.00 | 0.00-0.07 | N/A |
| Positive listening | 0.88 | 0.76-0.94 | $0.92 \pm 0.03$ |
| Positive signalling | 0.98 | 0.90-1.00 | $0.99 \pm 0.00$ |
| Both | 1.00 | 0.93-1.0 | $0.98 \pm 0.01$ |

Table 1: Both biases lead to consistent discovery of useful communication. We define a good run to be one with final average reward greater than 0.2. Averages are over 50 runs for each setting.

### 4.2 Treasure Hunt

We propose a new cooperative RL environment called *Treasure Hunt*, where agents explore several tunnels to find treasure [1]. When successful, both agents receive a reward of 1. The agents have a limited field of view; one agent is able to efficiently find the treasure, but can never reach it, while the other can reach the treasure but must perform costly exploration to find it. In the optimal solution, the agent which can see the treasure finds it and communicates the position to the agent which can reach it. Agents communicate by sending one of five discrete symbols on each timestep. The precise generation rules for the environment can be found in the supplementary material.

The agents used in this environment are Advantage Actor-Critic methods [28] with the V-trace correction [7]. The agent architecture employs a single convolutional layer, followed by a multi-layer perceptron. The message from the other agent is concatenated to the output of the MLP, and fed into an LSTM. The network's action policy, message policy and value function heads are linear layers.

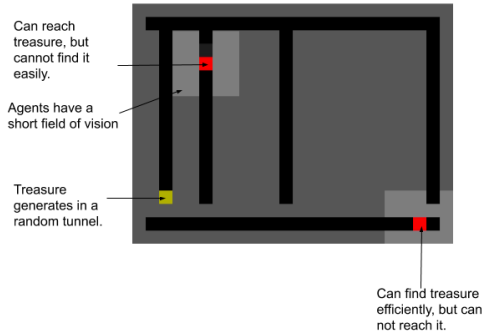

Figure 3: Treasure hunt environment.

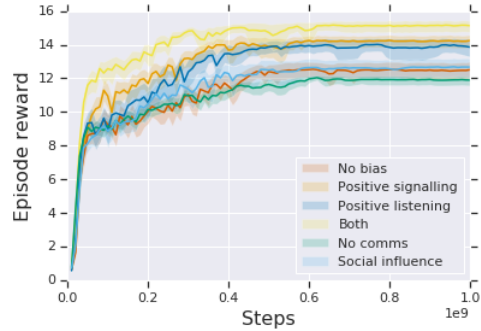

Figure 4: Positive signalling and listening biases leads to more reward.

| Biases | Proportion good | CI | Final reward (good runs) | Final reward |
|---|---|---|---|---|
| No bias | 0.28 | 0.18-0.42 | $14.67 \pm 0.29$ | $12.45 \pm 0.48$ |
| Positive signalling | 0.84 | 0.71-0.92 | $14.69 \pm 0.18$ | $14.22 \pm 0.36$ |
| Positive listening | 0.64 | 0.50-0.76 | $14.95 \pm 0.20$ | $13.94 \pm 0.44$ |
| Both | 0.94 | 0.84-0.98 | $15.41 \pm 0.14$ | $15.14 \pm 0.33$ |

Table 2: Proportion and average reward of good runs. Values are means over 50 runs with 95% confidence intervals, calculated using Wilson approximation in the case of Bernoulli variables.

| Run | Mean time (unmodified) | Mean visit time (modified) |
|---|---|---|
| Median | $100.6 \pm 14.7$ | $36.1 \pm 3.3$ |
| Best | $85.4 \pm 14.1$ | $41.3 \pm 7.9$ |

Table 3: Visit time to tunnel, with and without modified messages. Values are means over 100 episodes with 95% confidence intervals.

Our training follows the independent multiagent reinforcement learning paradigm: each agent is trained independently using their own experience of states and actions. We use RMSProp [13] to adjust the weights of the agent's neural network. We co-train two agents, each in a consistent role (finder or collector) across episodes.

The results are shown in Table 2. We find that biases for positive signalling and positive listening both lead to increased reward, and adding either bias to the agents leads to more consistent discovery of useful communication protocols; we define these as runs which get reward greater than 13, the maximum final reward in 50 runs with no communication. With or without biases, the agents still frequently only discover local optima - for example, protocols where the agent which can find treasure reports on the status of only one tunnel, leaving the other to search the remaining tunnels. This demonstrates a limitation of these methods; positive signalling and listening biases are useful for finding some helpful communication protocol, but they do not completely solve the joint exploration problem in emergent communication. However, among runs which achieve some communication, we see greater reward on average among runs with both biases, corresponding to reaching better local optima for the communication protocol on average.

We also ran experiments with the speaker getting an extra reward for influencing the listener, as in [15]. Here, we used the centralized model in [15], where the listener calculates the social influence of the speaker's messages, and the speaker gets an intrinsic reward for increasing this influence. We did not see a significant improvement in task reward, as compared to communication with no additional bias.

We analyze the communication protocols for two runs, which correspond to the two videos linked in[1]. One is a typical solution among runs where communication emerges; we picked this run by taking the median final reward out of all runs with both positive signalling and positive listening biases enabled. Qualitatively, the behaviour is simple – the finder finds the rightmost tunnel, and then reports whether there is treasure in that tunnel for the remainder of the episode. The other run we analyze is the one with the greatest final reward; this has more complicated communication behaviour. To analyze these runs, we rolled out 100 episodes using the final policies from each.

First, we relate the finder's communication protocol to the actual location of the treasure on this frame; in both runs, we see that these are well correlated. In the median run, we see that one symbol relates strongly to the presence of treasure; when this symbol is sent, the treasure is in the rightmost tunnel around 75% of the time. In the best run, where multiple tunnels appear to be reported on by the finder, the protocol is more complicated, with various symbols correlating with one or more tunnels. Details of the correlations between tunnels and symbols can be found in the supplementary material.

Next, we intervene in the environment to demonstrate that these communication protocols have the expected effect on the collector. For each of these pairs of agents, we produce a version of the environment where the message channel is overridden, starting after 100 frames. We override the channel with a constant message, using the symbol which most strongly indicates a particular tunnel. We then measure how long the collector takes to reach a square 3 from the bottom, where the agent is just out of view of the treasure. In Table 3, we compare this to the baseline where we do not override the communication channel. In both cases, the collector reaches the tunnel significantly faster than in the baseline, indicating that the finder's consistent communication is being acted on as expected.

## 5   Conclusion

We introduced two new shaping losses to encourage the emergence of communication in decentralized learning; one on the speaker's side for positive signalling, and one on the listener's side for positive listening. In a simple environment, we showed that these losses have the intended effect of easing the learning problem for the other agent, and so increase the consistency with which agents learn useful communication protocols. In a temporally extended environment, we again showed that these losses increase the consistency with which agents learn to communicate.

Several questions remain open for future research. Firstly, we investigate only fully co-operative environments; does this approach can help in environments which are neither fully cooperative nor fully competitive? In such settings, both positive signalling and positive listening can be harmful to an agent, as it becomes more easily exploited via the communication channel. However, since the losses we use mainly serve to ensure the communication channel starts to be used, this may be as large a problem as it initially seems. Secondly, the environments investigated here have difficult communication problems, but are otherwise simple; can these methods be extended to improve performance in decentralized agents in large-scale multi-agent domains? There are a few dimensions along which these experiments could be scaled – to more complex observations and action spaces, but also to environments with more than two players, and to larger communication channels.

## Footnotes

[1]Videos for this environment can be found at https://youtu.be/eueK8WPkBYs and https://youtu.be/HJbVwh10jYk.

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
