[Supplementary Material]

# Supplementary material for
# Biases for Emergent Communication in
# Multi-agent Reinforcement Learning

## A  Statistical methodology

All confidence intervals shown are $95\%$ confidence intervals. For confidence intervals of Bernoulli variables – the proportion of runs with reward above a certain threshhold – we use the Wilson approximation. For graphs depicting average performance over multiple runs, we first take the mean reward per run in 100 time windows over training. The interval shown is the $95\%$ confidence interval for this mean.

## B  Environment details

To generate a map for the treasure hunt environment, we:

1. Create a rectangle of grey pixels with height $18$ and width $24$.

2. Draw a black tunnel on the second row up, including all but the leftmost and rightmost pixels.

3. Draw a black tunnel on the second row down, including all but the leftmost and rightmost pixels.

4. Pick $4$ starting positions on the top horizontal tunnel for vertical tunnels. These are randomly selected among sets of positions which are all at least 3 pixels apart (so no tunnel is visible from another). Draw black tunnels from these positions to 2 pixels above the bottom tunnel.

5. Place the yellow treasure at the bottom of a random tunnel.

6. Place one agent uniformly at random in the top tunnel, and one uniformly at random in the bottom tunnel.

The episode length is $500$ timesteps. The agents have $5$ actions, corresponding to the $4$ directions and a no-op action. They can move in the black tunnels and onto the treasure, but not onto the grey walls. The agent observation is a $5 \times 5$ square, centered on the agent. When an agent moves onto the treasure, both agents receive reward $1$, and the treasure respawns at the bottom of a random tunnel.

The RGB values of the colors of the pixels in the observations are:

- Blue self: $(0, 0, 255)$.

- Red partner agent: $(255, 0, 0)$.

- Grey walls: $(128, 128, 128)$.

- Black tunnels: $(0, 0, 0)$.

- Yellow treasure: $(255, 255, 0)$.

## C    Treasure Hunt protocol

For the run with median final reward in the final experiments using both positive signalling and listening biases, Table 1 shows the frequency with which the treasure was actually located in each tunnel, given the symbol sent by the speaker. Table 2 shows the same information, for the best performing run.

| Symbol \ Tunnel | 1 | 2 | 3 | 4 |
|---|---|---|---|---|
| 0 | $0.12 \pm 0.01$ | $0.08 \pm 0.01$ | $0.35 \pm 0.03$ | $0.46 \pm 0.03$ |
| 1 | $0.36 \pm 0.02$ | $0.35 \pm 0.02$ | $0.22 \pm 0.01$ | $0.07 \pm 0.01$ |
| 2 | $0.06 \pm 0.01$ | $0.05 \pm 0.00$ | $0.14 \pm 0.01$ | $0.75 \pm 0.02$ |
| 3 | $0.32 \pm 0.01$ | $0.35 \pm 0.02$ | $0.24 \pm 0.01$ | $0.09 \pm 0.01$ |
| 4 | $0.27 \pm 0.01$ | $0.28 \pm 0.02$ | $0.24 \pm 0.01$ | $0.20 \pm 0.01$ |

Table 1: Distribution of treasure location depending on speaker's message for median run.

| Symbol \ Tunnel | 1 | 2 | 3 | 4 |
|---|---|---|---|---|
| 0 | $0.08 \pm 0.01$ | $0.12 \pm 0.01$ | $0.32 \pm 0.02$ | $0.48 \pm 0.02$ |
| 1 | $0.15 \pm 0.01$ | $0.31 \pm 0.02$ | $0.31 \pm 0.02$ | $0.22 \pm 0.01$ |
| 2 | $0.12 \pm 0.01$ | $0.15 \pm 0.01$ | $0.29 \pm 0.02$ | $0.44 \pm 0.02$ |
| 3 | $0.56 \pm 0.03$ | $0.19 \pm 0.02$ | $0.16 \pm 0.02$ | $0.09 \pm 0.02$ |
| 4 | $0.23 \pm 0.02$ | $0.32 \pm 0.02$ | $0.27 \pm 0.02$ | $0.18 \pm 0.02$ |

Table 2: Distribution of treasure location depending on speaker's message for best run.

## D    Network details and hyperparameters

### D.1    MNIST sums experiments

In the MNIST sums environment, the agent architecture used was from an existing MNIST classifier; we did not optimize this, as the goal was to investigate the effect of communication biases rather than to achieve optimal performance. This architecture is:

- A convolutional neural network, with 2 layers, which have 32 and 64 channels respectively. Both layers have kernel size 5, stride 1, and rectified linear unit (ReLU) activations. We use max pooling, with stride and kernel 2.
- One hidden linear layer with 1024 neurons, with ReLU activations.

We used the Adam optimizer [3], with a learning rate of 0.0003 and parameters $\beta_1 = 0.9$, $\beta_2 = 0.999$, $\epsilon = 10^{-8}$.

For the listener agent, the message is concatenated to the flattened output of the convolutional net before the hidden linear layer.

The final hyperparameters used for the 4 settings were:

To select the hyperparameters for the no bias setting, we performed a joint sweep over action and message entropy bonuses, consider a range of values from 0.0 to 0.3 for each. No values were found which improved over the no communication policy; the final values reported here are those which worked best in the other settings.

In the positive listening setting, we performed sweeps:

- Over the weight of $L_{pl}$, using values in $(0.01, 0.03, 0.1, 0.3)$.
- Over the entropy costs for messages and actions, using values $(0.01, 0.03, 0.1)$.

| Hyperparameter | No bias | Positive speaking | Positive listening | Both |
|---|---|---|---|---|
| Batch size | 32 | 32 | 32 | 32 |
| Action policy entropy bonus | 0.03 | 0.03 | 0.03 | 0.03 |
| Message policy entropy bonus | 0.03 | 0.0 | 0.03 | 0.0 |
| Target message entropy $\mathcal{H}_{target}$ | N/A | 1.0 | 0.03 | 1.0 |
| Weight of $L_{pl}$ | N/A | N/A | 0.01 | 0.01 |
| Weight of $L_{ps}$ | N/A | 0.1 | N/A | 0.1 |
| Weight of $L_{ce}$ | N/A | N/A | 0.001 | 0.001 |
| $\lambda$ for $L_{ps}$ | N/A | 3.0 | N/A | 3.0 |

Table 3: Hyperparameters for final MNIST experiments.

- Over the weight of $L_{ce}$, using values of $(0.0, 0.001, 0.01, 0.1)$; aside from 0, which unsurprisingly produced worse results, there was no significant difference in the results of these runs.

In the positive speaking setting, we performed sweeps:

- Over the weight of $L_{ps}$ and $\lambda$, using values in $(0.01, 0.03, 0.1)$ for $L_{ps}$, and $(0.01, 0.03, 0.1)$ for the product $\lambda L_{ps}$.

In the setting with both biases, we ran no additional sweeps, simply combining the hyperparameters from the best runs with positive speaking and positive listening.

For all hyperparameter sweeps, we ran 5 runs, and picked the setting with the highest average final reward. For the final sets of hyperparameters, we then ran 50 runs.

## D.2 Treasure Hunt sums experiments

In our experiments, we use 32 parallel environment copies for the Advantage Actor-Critic algorithm [4] with the V-trace correction [1]. The two agents have the same architecture, which consists of:

- A single convolutional layer, using 6 channels, kernel size of 1 and stride of 1.
- A multi-layer perceptron with 2 hidden layers of size 64.
- An LSTM, with hidden size 128.
- Linear layers mapping to policy logits for the action and message policies, and to the baseline value function.

The message from the other agent is concatenated to the flattened output of the convolutional net before the hidden linear layer.

We used the RMSProp optimizer [2] for gradient descent, with a initial learning rate of 0.001, exponentially annealed by a factor of 0.99 every million steps. The other parameters are $\eta = 0.99$, and $\epsilon = 10^{-6}$.

The final hyperparameters used for the 5 settings were:

| Hyperparameter | No comms | No bias | PS | PL | Both |
|---|---|---|---|---|---|
| Batch size | 16 | 16 | 16 | 16 | 16 |
| Action policy entropy bonus | 0.006 | 0.006 | 0.006 | 0.006 | 0.006 |
| Message policy entropy bonus | N/A | 0.0 | 0.0 | 0.0 | 0.0 |
| Target message entropy $\mathcal{H}_{target}$ | N/A | N/A | 0.8 | N/A | 0.8 |
| Weight of $L_{pl}$ | N/A | N/A | 0.003 | N/A | 0.003 |
| Weight of $L_{ce}$ | N/A | N/A | N/A | 0.01 | 0.01 |
| Weight of $L_{ps}$ | N/A | N/A | 0.001 | 0.001 | N/A |
| $\lambda$ for $L_{ps}$ | N/A | N/A | 3.0 | N/A | 3.0 |

Table 4: Hyperparameters for final Treasure Hunt experiments.

In the no communication setting, we performed sweeps:

- Over the entropy costs for actions, using values $(0.001, 0.003, 0.006, 0.01)$.
- Over the sizes of the MLP layers, using values $(32, 64, 128)$.
- Over the sizes of the LSTM hidden size, using values $(64, 128)$.

We then fixed these parameters for the other settings.

In the positive listening setting, we performed sweeps:

- Over the weight of $L_{pl}$, using values in $(0.001, 0.003, 0.01)$.
- Over the weight of $L_{ce}$, using values of $(0.0, 0.001, 0.01)$.
- Over the message policy entropy bonus, using values of $(0.0, 0.001, 0.003, 0.006, 0.01)$.

In the positive speaking setting, we performed sweeps:

- Over the weight of $L_{ps}$ and $\lambda$, using values in $(0.001, 0.003, 0.01)$ for $L_{ps}$, and $(0.001, 0.003, 0.01)$ for the product $\lambda L_{ps}$.
- Over the target entropy $\mathcal{H}_t$, using values in $(0.4, 0.8, 1.6)$.

In the setting with both biases, we ran no additional sweeps, simply combining the hyperparameters from the best runs with positive speaking and positive listening.

For all hyperparameter sweeps, we ran 5 runs, and picked the setting which exceeded the no-communication baseline most frequently, terminating runs early if the result was clear. For the final sets of hyperparameters, we then ran 50 runs.