[Reviews · NeurIPS 2019]

Reviewer 1



The authors present two losses for improving emergent communication, addressing concerns laid out in previous work (Lowe 2019). One is the concern that the speaker agent may be communicating generic messages and not ones relevant to the particular sitation. A loss here encourages the agent to send messages that are correlated with their observation, based on maximizing mutual information between them. A second concern is that the listener agent may not be conditioning their behavior on the communication, and in this case an extra loss Both constraints are intuitive, and phrasing them as losses doesn't seem to be particularly challenging. However, as with many issues in emergent communication, such a judgement may gloss over hidden difficulties in a complex optimization problem, and this appears to be the case here, requiring some non-obvious sidestepping to provide losses with better convergence. We're not supplied detailed analysis of what failed, but at face value, having the losses formulated in a way that has shown to be useful for optimization is a useful contribution to all researchers working in this area (regardless of the conceptual simplicity of the idea). There are some concerns about the long-term usefulness of these biases, especially of the positive listening loss. While this bias may make sense in simple optimization problems, in reality, a change in agent strategy is not necessarily indicative of listening, nor is the lack of such change an indication of a poor listener. I was a bit surprised not to see any discussion about how realistic these loss-motivating assumptions are in the bigger picture. But this is a minor concern, as we are still very much in the realm of toy examples. As for the evaluation, overall strong improvements shown for either of the losses over the baseline, and for both losses used together. I thought overall the experiments were well chosen -- one where the RL environment was reduced to a trivial degree, and one which closer reflects the types of domains used in related work. Both seem well-controlled. The baselines seemed a bit too outmatched here and raises the question of whether or not there are some optimization tricks that could have been utilized, rather than setting up difficult problems where the baseline is inclined to fail. But assuming "worst case", being presented with such a problem where no curriculum learning or reward shaping options are available, the provided methods show improvements both in the percent of good strategies reached, and the rewards collected from such policies. The paper is also very clearly written, and so while the technical difficulty of the methods presented here is not high, nor are they particularly novel, the paper represents a concise contribution of effective optimization strategies, broadly applicable to the emergent communication literature, and is certainly ready for acceptance at some venue. Perhaps its greatest shortcoming is simply whether it meets the NeurIPS bar in terms of its novelty/technicality. Some in-line comments: L 127: (2) is formulated with trajectories but it wasn't clear if this more general interpretation was useful. Does this have a noticable effect over a single state? In general, states and trajectories feel a bit mixed together throughout S3/3.1. L 139/143.2: The loss notation here changes from L_S to L_ps L 143/Alg 1: Inconsistent period usage L 143/Alg 1: Lines 3-12 should be clearly marked as documentary/comments L 143.15: what is the purpose of having the hidden state passed to p_t^i? In Alg 2 this makes sense as it is required for conditioning on future roll-outs, but in the positive signalling loss too? Fig 2: these are compelling plots but presumably the data samples are chosen at random and from the full label size. The losses are useful compared to baselines without, but have you considered approaching the problem from the data side? I would be interested to know if simplifying the problem (in terms of label size), and gradually increasing it / setting a curriculm, would have a positive effect on the baselines. It appears the baselines are just setup too poorly here, with little effort given to "make them work". L 222: is Tab 1 referenced? L 242: typically not a context for a semi-colon

Reviewer 2



Summary: The paper proposes the use of two extra term losses that encourage positive signalling and listening in multi-agent reinforcement learning settings where agents have access to a communication channel. The authors show that this leads to agents that learn to more robustly use the communication channel (across different runs) compared to agents that are not trained with these extra losses (or intrinsic rewards) or only use one of these terms. Strengths: The paper is generally clear and well-structured The paper addresses an important problem in MARL and particularly, attempts to tackle the more challenging and realistic setting of decentralized training and execution of the agents, without direct access to other agents’ parameters, rewards, actions, states, or internal beliefs and preferences. I like the various discussions throughout the paper that provide explanations and intuitions for the behavior of the agents or the effects or different losses on their performance (e.g. section 4.2). The paper appears to be technically correct. I also appreciated the fact that the authors openly acknowledged some of the limitations of their method (e.g. section 4.2) Weaknesses: The paper could benefit from comparisons against stronger and more diverse baselines, such as the method proposed by Jaques et al. 2019 (which also uses decentralized training and execution), Foerster et al. 2016 (e.g. RIAL which is concerned with the same setting), or Sukhbaatar et al. 2016 The paper lacks a large number of references to related work. The auxiliary losses proposed are a form of reward shaping for improving MARL algorithms (e.g. prosociality, curiosity, empowerment, optimistic Q-learning etc.), on which there exists a large body of work which is not discussed in the paper. Some examples of related papers are: Peysakhovich & Lerer (2018), Devlin et al. (2014), Foerster et al. (2018), Oudeyer & Kaplan (2006), Oudeyer & Smith (2016), Forestier & Oudeyer (2017). While the simplicity of the chosen tasks helps to understand in greater detail what the agents’ behavior, I think the paper requires more evaluation on more complex tasks. In particular, I think many readers would be interested in a discussion of the scalability of this method to more than 2 agents. Moreover, the number of symbols used in the communication channel is quite small and it would be good to see how the method performs as the vocabulary increases. Some of the results don’t seem that impressive. While the proposed method allows agents to use the communication channel more often than the alternatives, it does not improve performance by a very significant amount, when comparing the runs that make use of communication. Numerous details about the algorithm used for optimization are missing. In particular, what is the total loss used for optimization in both tasks? How does that relate to RIAL or other previously proposed MARL algorithms? Other Comments: It would be good to provide a plot or table with the final reward across all runs. I expect that number to show their method as obtaining much better rewards on average, given the larger proportion of runs in which the communication channel is actually being used by the agents. It is not very clear to me why the agents still learn suboptimal communication protocols even when learning to use the communication channel. Have the authors tried to use a more powerful RL algorithm such as PPO, SAC or A3C? Why did the authors decide to use a multi-step version of the CIC algorithm? It would be useful to provide an ablation study in which they also compare against the single step version. There are some missing details when going from equation (5) to (6) that should at least appear in the supplementary material. The proposed method is novel (as far as I can tell) and the problem is of wide interest, so I believe a stronger version of this submission would be of interest to the community. However, I do believe the paper as it stands right now requires more empirical evaluation against stronger baselines and on more complex environments. ------------- UPDATE: I have read the rebuttal and the other reviews. I appreciate the fact that the authors took into account the feedback, clarified some parts of the paper, and promised to add more comparisons with prior work, analysis of the learned communication protocols and relevant references. Their rebuttal helped me understand why they made certain decisions and what the scope of the paper is. I now lean towards acceptance and I have updated my score to 6.

Reviewer 3



One of the most important open problems in emergent communication is the problem of discovery: how can two randomly-initialized agents stumble upon a communication policy that transmits useful information and helps solve a given task? Many previous approaches have used gradient information passed from the listener to the speaker in order to improve the learning of communication protocols, however this has some drawbacks. This paper provides an alternate approach, which is to add additional rewards to the decentralized, discrete-message framework to encourage communication. The authors show that these biases improve performance on two separate communication games. I quite like this paper. It is well-written, the methodology makes sense, and the results clearly show that the proposed biases (which incentivizes the ‘speaker’ agent to send messages correlated with its state, and incentivizes the ‘listener’ agent to take the speaker’s messages into account when acting) improve performance. The environments tested (MNIST adding and the new ‘treasure hunt’ environment) are fairly simple, but in my opinion interesting enough to show the benefit of the proposed approach. The implementation of these biases is also non-trivial, and the paper walks through how they are derived in detail. My main concern about the paper is the comparison to Jaques et al. (2018). As the authors mention, incentivizing ‘positive listening’ was previously investigated in Jaques et al. by giving the *speaker* a reward for sending messages that had a large influence on the listener. In contrast, this paper rewards the *listener* for being influenced by the speaker. Given that these two objectives are fairly similar, it is surprising to me that the paper doesn’t compare to the influence formulation of Jaques et al. I think the experimental results would be much stronger if this comparison was included, along with an explanation of the pros / cons of each approach. Finally, given that a similar approach to the one in this paper was taken by Jaques et al., the novelty of the paper is more limited. Overall, I would recommend this paper for acceptance if there was a more extensive comparison to Jaques et al. For now, I will give this paper a weak accept, but am willing to update my review accordingly. Small comments: - L166: We use is the -> We use the - L172: is new -> is a new - L176: the results protocols -> the resulting protocols ------------------------------------------------------- After reading the author's rebuttal, I will also increase my score by 1 point, from a 6 to a 7. My main concern was the comparison to the similar formulation from Jaques et al., and I'm convinced by the authors assertion that this does not work in the MNIST setting. I'm hopeful to see this comparison in the final version. My sole remaining concern is how Jaques et al.'s method will compare in the Treasure Hunt game (the authors state that they 'will run this'), and whether this result will be included even if it shows their method does worse. Despite this concern, I now reside more firmly in the 'accept' camp.

[Author Response · NeurIPS 2019]

We thank all the reviewers for their exceptionally careful reading of the paper, and their very helpful comments.

Two reviews commented that comparisons to more other algorithms would be helpful. The algorithm of clearest
relevance here is that introduced in Jaques et al. (2018). This algorithm can be seen as introducing a bias encouraging
the speaker to increase positive listening. We already note in the paper that this bias did not help with solving the
MNIST task, but will emphasize this point. Further, we elaborate the discussion of this in three ways. Firstly, we will
display the results in full, including the analogue of Figure 2, rather than simply state that the algorithm from Jaques
et al. does not outperform the no-communication baselines. Secondly, we will run a comparison using the bias from
Jaques et al. on Treasure Hunt. Finally, we will explain why we believe the bias from Jaques et al. is unhelpful in this
setting. In short, this is because for a fixed listener, the speaker policy which optimizes positive listening has no relation
to speaker's input. Thus this bias does not force the speaker to produce different message for different inputs, and so
does not increase the learning signal for the listener. This is true when the observations of the speaker and listener are
independent (such as in the MNIST task). In that case, the expected positive listening for the speaker is a function only
of its message, not its observation. And so the best policy to increase this expectation doesn't depend on the observation.
In Treasure Hunt, it's not quite true that the speaker should ignore its state to increase positive signaling - it should only
use the state to deduce the listeners state, and therefore how it might best be influenced.

The other suggested baselines are RIAL from Foerster et al. 2016 and Sukhbaatar et al. (2016). RIAL is very similar
to the baseline used in the paper; the difference is in the underlying RL algorithms (DQN in RIAL, and A3C here).
Sukhbaatar et al. (2016), and other approaches we are aware of, use a differentiable model of communication. While
also interesting, this is not the domain we are examining here, so we have not run these baselines.

There were two suggestions for improvements to the related work section; other emergent communication work (e.g.
COMA from Foerster et al. (2017), Sukhbaatar et al. (2016)), and the extensive literature on auxiliary losses, particularly
in MARL. We agree with both, and we updated the paper accordingly.

Two of the reviews expressed a desire to see these algorithms tried on more complex tasks, which we agree is an
important direction. However, we believe that it is a novel contribution to achieve emergent communication in any
many-step RL environment with non-differentiable communication channels. We therefore think that these methods
represent an important advance, and are likely to contribute to solutions of harder communication problems, even if
more advances are needed. We updated the discussion in the paper to consider the possible scaling of the algorithm in
more detail - adding discussion of $n > 2$ agents, and discussing the implications of larger communication channels.

Reviewer 1 asks two related questions; why (2) is formulated with trajectories, and why line 12 of algorithm 2 includes
the speaker's hidden state. This is because the message policy is recurrent (which is necessary for a good protocol in
Treasure Hunt, due its partial observability); so the calculation of the message policy needs the hidden state, and the
natural formulation of the mutual information in positive speaking is with the trajectory rather than the state.

Reviewer 1 is correct to point out that the baselines in the MNIST task are not heavily optimized; with a curriculum
approach as suggested, we would not be surprised if this task could be solved (at least sometimes) without the biases we
use. However, the main purpose of this task is to examine how these biases affect learning, we don't think optimising
the baselines is as important as in other contexts.

Reviewer 1 comments that runnable code or message information would be helpful. In an appendix, we will provide a
wider variety of message protocols, summarizing them in a similar way to those examined in the paper.

Reviewer 2 comments that it would be good to provide a plot or table with the final reward across all runs. We will add
this to Table 2. As the reviewer notes, this will have a significant difference between our methods and the baseline, due
to the increased rate with which the agents learn to communicate with the biases.

Reviewer 2 asks why the agents fail to achieve optimal performance, and notes that the performance is little better than
the baseline conditional on communication happening. The answer to this is in Section 4.2; these methods are primarily
aimed at getting communication to emerge in the first place, rather than at reaching the global optimum protocol. There
is still much work to be done in joint exploration in emergent communication.

Reviewer 2 asks about the total loss used. The loss is the usual loss for the RL algorithm being used (REINFORCE for
the MNIST problem, and A3C for Treasure Hunt), plus the losses we outline. We will clarify this in the paper.

Reviewer 2 asks about why we use multi-step, rather than single-step, CIC. Intuitively, we expect messages from several
steps previously to be relevant to the listener's decision. Empirically, we found the multi-step version to be superior; we
state this in the text and provide an ablation study for this (perhaps in an appendix, for want of space).

All minor corrections to notation and spelling are gratefully received, and we will make them for the next version. We
will also improve the legibility of Figure 2, and add a more detailed explanation of Equation 6 in the supplementary
material.

[Meta-Review · NeurIPS 2019]

This paper proposes simple but potentially quite significant ways to encourage useful emergent communication in MARL. The paper is well written with a nice empirical analysis. The reviewers had some concerns about the coverage of related work and the empirical comparison to Jacques et al. but it seems easily feasible to address these in the final version. The authors are strongly encouraged to add to the final version a comparison to Jacques et al. in Treasure Hunt, regardless of whether the results favour the proposed method.